# Is It Possible to Find an Antimicrobial Peptide That Passes the Membrane Bilayer with Minimal Force Resistance? An Attempt at a Predictive Approach by Molecular Dynamics Simulation

**DOI:** 10.3390/ijms23115997

**Published:** 2022-05-26

**Authors:** Ilya V. Likhachev, Nikolay K. Balabaev, Oxana V. Galzitskaya

**Affiliations:** 1Institute of Protein Research, Russian Academy of Sciences, 142290 Pushchino, Russia; ilya_lihachev@mail.ru; 2Institute of Mathematical Problems of Biology RAS, Keldysh Institute of Applied Mathematics, Russian Academy of Sciences, 142290 Pushchino, Russia; balabaev@psn.ru; 3Institute of Theoretical and Experimental Biophysics, Russian Academy of Sciences, 142290 Pushchino, Russia

**Keywords:** antibacterial peptides, MD simulation, membrane, penetration

## Abstract

There is still no answer to the mechanism of penetration of AMP peptides through the membrane bilayer. Several mechanisms for such a process have been proposed. It is necessary to understand whether it is possible, using the molecular dynamics method, to determine the ability of peptides of different compositions and lengths to pass through a membrane bilayer. To explain the passage of a peptide through a membrane bilayer, a method for preparing a membrane phospholipid bilayer was proposed, and 656 steered molecular dynamics calculations were carried out for pulling 7 amyloidogenic peptides with antimicrobial potential, and monopeptides (homo-repeats consisting of 10 residues of the same amino acid: Poly (Ala), Poly (Leu), Poly (Met), Poly (Arg), and Poly (Glu)) with various sequences through the membrane. Among the 15 studied peptides, the peptides exhibiting the least force resistance when passing through the bilayer were found, and the maximum reaction occurred at the boundary of the membrane bilayer entry. We found that the best correlation between the maximum membrane reaction force and the calculated parameters corresponds to the instability index (the correlation coefficient is above 0.9). One of the interesting results of this study is that the 10 residue amyloidogenic peptides and their extended peptides, with nine added residue cell-penetrating peptides and four residue linkers, both with established antimicrobial activity, have the same bilayer resistance force. All calculated data are summarized and posted on the server.

## 1. Introduction

Membrane-active peptides in nature perform many important functions, acting as antimicrobial agents, channels, transporters, or hormones [1]. Cell-penetrating peptides (CPPs) are capable of targeting various types of cargoes across the cell membrane, including drugs and nanoparticles. The effectiveness of pharmaceutical treatments is often reduced due to insufficient membrane permeability, which prevents drugs from reaching their specific intracellular targets.

An important physical characteristic of any membrane system is the ordering of bonds in the hydrocarbon chains of phospholipids. A significant part of the lipid molecules that make up natural membranes are unsaturated phospholipids. In membrane systems, they play the most important functional role [2].

The incorporation of a hydrophobic segment of peptide or membrane protein into a bilayer can occur spontaneously and mainly due to a hydrophobic effect. Anionic lipids, which make up about 20% of biological membranes, are a source of electrostatic attraction and ensure the binding of proteins (peptides) to membranes [3].

There is still no answer to the question about the mechanism of penetration of antimicrobial peptides (AMPs) through the bilayer of a membrane. Currently, work continues on the study of the penetration of peptides through the membrane [4]. The incorporation of the peptide into the lipid bilayer is triggered by a drop in pH, and its release from the bilayer is triggered by an increase in pH. Like phospholipids, peptides can be incorporated into the membrane. This process is described in [5]. Maximum injection pressure (MIP) was used to assess the degree of penetration of proteins and peptides into lipid monolayers [6].

The interactions of GF-17 derived from the antimicrobial peptide LL-37 using bacterial membranes is described in [7]. Spontaneous membrane-translocation adsorption of peptides has the ability to directly penetrate cell membranes [8]. This problem is also relevant for the translocation of the HIV-1 TAT peptide and conjugated gold nanoparticles through lipid membranes [9]. Synthetic interference peptides containing a cell-penetrating peptide are described in [10]. In our work, we also study the penetration of a hybrid peptide containing a fragment of the HIV-1 TAT peptide through the membrane using steered molecular dynamics simulations.

One approach to incorporating a peptide into a membrane bilayer is described in [11]. The correlation between peptide–lipid interaction and penetration efficiency is described in [12]. Studies have been carried out on the involvement of membrane peptides in cancer processes [13]. There are studies on the activation of neuraminidase-1 by transmembrane peptides, in particular, by the method of molecular dynamics (MD).

The molecular dynamics of peptides at high pressure is described in [14]. The penetration of peptides through the membrane leads to a local increase in pressure in the membrane. This can be neglected if the membrane is intact. The effect on the mechanical properties of the cell membrane is described in [15].

A review of methods for studying the passage of peptides through a membrane is described in [16]. The process of peptide penetration into the membrane itself can be considered as diffusion [17]. The study in [18] shows the use of the molecular dynamics method to study the mechanism of the absorption of peptides by the cell membrane due to interaction with phosphate heads. The work in [19] is based, again, on the diffusion approach, using Fick’s first law. In this work, we use steered molecular dynamics, i.e., we do not expect spontaneous penetration of peptides into the membrane.

As described in [20], the MD-simulation method is not the only method used to study the penetration problem. Spectroscopic and zeta potential methods are also used for this task.

In this work, we studied the passage of various peptides across the 1-palmitoyl-2-oleoyl-sn-glycero-3-phosphatidylcholine (POPC) membrane. The preparation of such membranes for MD calculation is described in [21]. Among the peptides studied in this work, we managed to identify a peptide that passes through the membrane more easily than other peptides. Moreover, the addition of a cell-penetrating peptide (HIV-1 TAT peptide) does not greatly affect its passage through the membrane. The simulation protocol of the present work somewhat resembles the protocol of work [22]. We use bilayer relaxation, along with the peptide in the NPT ensemble, and then move on to the NVT ensemble. The main difference is that we are dragging the entire peptide through the membrane, rather than stopping at the middle (5 Å after the middle of the membrane, to be exact). The phospholipid bilayer holds on to our common center of mass, and the peptide is dragged through both the atom and the center of mass.

## 2. Results

### 2.1. Studied Amyloidogenic Peptides with Antibacterial Potential and Homo-Repeats

Of the seven amyloidogenic peptides studied in this work, two, after experimental verification, turned out to exhibit antibacterial activity: peptide 1, RKKRRQRRRGGAGVTDFGVFVEI, (R23I)^T^, peptide 6, VVEGTVVEVT, (V10T)^T^, and peptide 7, VTDFGVFVEI, (V10I)^T^ [23] (see Table 1). The last is an amyloidogenic peptide from the *Thermus thermophilus* ribosomal S1 protein. Peptide 1 is a modification of peptide 7, created by adding a cell-penetrating peptide (HIV-1 TAT peptide) and a linker (GlyGlyAlaGly) to it. In total, we have a peptide with a length of 23 amino acid residues. The addition of CPP only enhanced the antimicrobial properties of the amyloidogenic peptide [23]. From the point of view of the penetration of an amyloidogenic peptide and its elongated analog, it is interesting to see not only its antimicrobial potential, but also to compare its ability to penetrate the membrane with a 10-membered peptide 7 (P7). Will the added cell-penetrating peptide inhibit the process itself, compared to a short peptide? Moreover, it is interesting to compare representative arginine-rich cell-penetrating peptides, such as TAT peptide, with polyarginine.

The ability of the predicted amyloidogenic peptides P1–P7 to aggregate and form amyloid-like fibrils was first checked by the fluorescence intensity of thioflavin T and electron microscopy (Table 1) [24].

In addition to the peptides themselves, corresponding to the amyloidogenic regions of the ribosomal S1 proteins from *T. thermophilus* and *E. coli*, we considered additional monopeptides, which are homo-repeats consisting of 10 residues of the same amino acid. The amino acids that make up our peptides were selected based on the size of the amino acid itself passing through the membrane bilayer: Poly (Ala), Poly (Leu), Poly (Met), Poly (Arg), Poly (Glu).

### 2.2. Construction of Membrane Phospholipid Bilayer

The method for preparing the membrane phospholipid bilayer is described in the Methods section. The construction of 8 copies of the phospholipid molecule (Figure 1) along the OY axis, and the construction of 8 copies of the resulting system along the OZ axis has been done in such a way that one layer includes 64 molecules (8 × 8) of phospholipids (Figure 2). After obtaining the bilayer, the studied peptide was added to it in an extended conformation (Figure 2). Then, the required amount of water was added to carry out calculations in periodic boundary conditions, with a parallelepiped as a calculation cell (Figure 3). Periodic boundary conditions were introduced in order to simulate an infinite layer of the membrane.

### 2.3. MD Simulations

A series of numerical experiments (8 independent realizations) were carried out with each peptide to push it through the membrane at constant rates of 0.1, 0.05, and 0.01 Å/ps by the terminal amino acid residue and by the center of mass. Thus, for each peptide, 48 trajectories with the basic peptides (P1–P7) + 24 trajectories with CPP were obtained. In total, for all peptides, 656 steered molecular dynamics experiments were carried out (8 × 6 × 8 main + 22 × 8 auxiliaries + 2 × 6 × 8 controls). For P1 and P4 peptides, a control experiment was performed: the MD calculations were repeated, and the qualitative result was the same. All results can be found at http://lmd.impb.ru/protres_membrane/ (accessed on 25 May 2022).

We were interested in the force response of the peptide when it was pulled through the lipid bilayer. A typical force response when the P2 peptide is pulled by the atom at a constant rate of 0.1 Å/ps is shown in Figure 4. As expected, the force response of the peptide and membrane is opposite in sign and approximately equal in magnitude.

In each MD experiment, the maximum force response of the peptide was calculated. The average values of force maxima for 8 realizations when the peptide was pulled over the terminal atom and the center of mass are presented in Table 2.

During relaxation, the peptides became compact, since this state is energetically more favorable (Figure 5). The resistance area of the compact structure is larger. When a peptide is pulled through the bilayer by an atom, its molecule becomes linear again. For all 8 realizations of all series of the experiments, the maximum value of the force was found (Figure 6). It turns out that the force reaction of the system when the peptide is pulled out by the terminal atom is always lower than when it is pulled out by the center of mass, due to the fact the area of resistance to the peptide is smaller in the first case (Table 2, Figure 6).

To obtain more reliable results, each computational experiment was repeated 8 times. For peptides 2 and 4, an additional series of independent computational experiments (48 for each peptide), was performed, including the relaxation stage. Calculations showed that the difference between the maximum forces in these experiments is minimal.

### 2.4. Division of Peptides into Groups According to the Force Reaction of the Membrane

Despite the fact that 11 peptides have the same length, the strength of the membrane reaction is different for all. This means that the force resistance varies, depending on the peptide sequence. What is the driving force for the membrane penetration of peptides in the POPC membrane in our system? It is most likely explained by the hydrophobic interaction of a peptide with the POPC membrane because peptides with higher hydrophobicity appear to be advantageous for lipid membrane penetration (Table 2). Force resistance depends on at least two factors: (1) partial charges in atoms; (2) the amino acid resistance area. Having calculated several parameters for peptides, (Z) is the value of charge at pH 7, pI is the theoretical value of isoelectric point, μHn is the normalized hydrophobic moment, Hn is the normalized hydrophobicity, PD is the penetration depth [26,27], α is the aliphatic index, and II is the instability index [28]; we found that the best correlation between the maximum membrane reaction force and the calculated parameters corresponds to the instability index (the correlation coefficient is above 0.9 in the case when the peptide is pulled by the center of mass, see Appendix A).

If the peptides are divided into clusters according to the force reaction of the membrane when the molecule is pulled by the terminal atom, then we can distinguish a class of strong reaction (peptide 2, Poly (Arg) and CPP), a class slightly below a strong reaction, (Poly (Glu), peptide 1 and peptide 5), a class of weak reaction (Poly (Ala)), and a middle class, comprised of the remaining peptides that cannot be ranked.

The results of the steered MD simulations with peptide pulling by the center of mass largely repeat the results of pulling by the terminal atom. However, in the results of the steered MD simulations with pulling by the atom, a greater similarity is observed in statistically independent realizations with the same peptide. We believe that in experiments with pulling by the center of mass, the result largely depends on the conformation of the peptide.

Comparison of the MD results (in general) shows that pulling the peptide by the center of mass causes a more pronounced force response, definitely due to the compactization of the peptide. This conclusion correlates well with the result, where it was shown that the main force reaction occurred precisely when a molecule was inserted into the membrane.

If we consider the effect of the pulling velocity on the reaction force of the membrane, then, as expected, a high velocity causes a large force reaction. Among the unexpected results, it is worth noting the statistical significance of MD simulations with a high pulling rate (0.1 Å/ps). It was originally though that a lower velocity would give more accurate results; but it turned out that all simulations have approximately the same accuracy. Therefore, to save computer time, computational experiments at low rates (0.01 Å/ps) were not performed for some monopeptides and lipids.

### 2.5. Phospholipid Pulling

Numerical experiments were carried out to obtain additional information about the stability of the phospholipid layer. The peptide from the same layer was pulled by the center of mass at a constant rate of 0.1 Å/ps towards the head (forward lipid pulling experiment) and towards the tail (reverse lipid pulling experiment). The membrane itself was fixed by the center of mass.

When stretching the phospholipid, the membrane exerts practically no force effect. The force reaction of the next membrane layer, under periodic boundary conditions, is much higher than the reaction of the layer from which the phospholipid is pulled out (Figure 6A). This is despite the fact that there is one less phospholipid in the membrane layer and the cavity, which is quickly closing due to lateral pressure.

This conclusion correlates with the force reactions of the membrane layer to the pulling of peptides. When the peptide is inserted into the membrane, it overcomes a more pronounced resistance of the membrane than when passing through it.

## 3. Materials and Methods

### 3.1. Description of the Task

The task consists of the following steps:(1)Construction of phospholipid bilayers with peptides;(2)Carrying out MD experiments on peptide pulling (across a bilayer) at constant rates of 0.1, 0.5, and 0.01 Å/ps
(a)by terminal atom (8 numerical experiments for each peptide),(b)by the center of mass of peptide;(3)Collection and processing of statistical data for all experiments.

### 3.2. Software and Md-Simulation

PUMA [29,30,31,32,33,34] supports operation under periodic boundary conditions in NPT and NVT ensembles. PUMA-CUDA supports operation in periodic boundary conditions in an NVT ensemble; however, it has a high performance due to various parallel programming technologies (parallel work on multi-core systems with shared memory, and with distributed memory, as well as work on graphics accelerators). The AMBER [35] forcefield was used, as well as the TIP3P [36] water model.

The task of modeling molecular dynamics is to integrate the equations of motion of a system of *N* interacting material points (atoms), whose motion is described by classical Newton’s equations:(1)$$m_{i}\cdot \ddot{r_{i}}=F_{i}\equiv -\frac{\partial U}{\partial r_{i}}$$
where i = 1..N,
(2)$$U\equiv U\left(r_{1},\ldots ,r_{N}\right)=U_{valency\;bonds}+U_{valency\;angles}+U_{torsion\;angles}+U_{\mathrm{planar}\;\mathrm{group}\;\mathrm{energy}}+U_{\mathrm{van}\;\mathrm{der}\;\mathrm{waals}\;\mathrm{energy}}+U_{coulomb\;energy}$$

Computationally, the problem is reduced to finding the energy and the forces, and integrating the equations of motion, along with taking into account external influences and boundary conditions. A collisional thermostat is used to simulate the system at a given temperature [29,30].

In our MD-experiments, the numerical integration step is 0.001 ps; the cut-off radius is 10.5 Å for non-valent interactions; the temperature is 350 K (we use a collisional thermostat). In the NPT-ensemble, the reference pressure was 0.1 MPa.

### 3.3. Construction of Phospholipid Bilayers with Peptides

The phospholipid molecule shown in Figure 1 was used as the basis for constructing the membrane phospholipid layer.

The construction of the membrane layer was achieved by using the following affine transformations:Construct 8 copies of the phospholipid molecule (multiplication) along with the OY axis (directed upwards relative to the plane of the paper).Construct 8 copies of the resulting system (multiplication) along the OZ axis (perpendicular to the plane of the sheet of paper). Thus, one layer of phospholipids, 8 × 8 = 64 molecules, was obtained.Construct a copy of the resulting system, and then its rotation by 180 degrees around the OY axis, placing the resulting layer tail to tail (Figure 2). Reflection is not suitable in this case, since it will change the topology of the molecules.Fill the system with a peptide with a given amino acid sequence, constructed using the PyMol molecular constructor. The first amino acid residue is preceded by a hydrogen molecule, and the last is preceded by an OH group covering the free valences. The resulting peptide is shown in Table 1.To simulate real conditions, a solvent (water) is added to the system. The number of molecules corresponds to the possibility of calculating molecular dynamics under periodic boundary conditions, with a parallelepiped as a calculation cell (Figure 3).The relaxation of one system is carried out in the PUMA software complex [29,30] in the NPT ensemble at a constant pressure of 1000 bar to clarify the computational cell size (volume).The relaxation of each system (with different peptides) is carried out in the PUMA-CUDA software package for 1 ns in the NVT ensemble, with calculation of the computational cell size of those obtained in step 6.

### 3.4. Force Effects

There are two ways of pulling the peptide through the membrane layer at a constant velocity: by the terminal (heavy) atom, and also by the center of mass.

Constant velocity refers to the constant velocity of the wall with an attached spring, whose other end is connected to the terminal atom of the peptide, or its center of mass.

The force acting on the terminal atom is *F* = *k*Δ*x*, where *k* is the stiffness of the spring, and Δ*x* is the distance of the spring from the atom.

When a peptide is pulled over the center of mass, a force is applied to each atom of the peptide, proportional to its mass:*F_i_* = *k*∙∆*x m_i_*/(∑*m_j_*)(3)
where *m_i_* is the mass of the *i*-th atom of the peptide; ∑*m_j_* is the total mass of the peptide.

To avoid displacement of the phospholipid layer according to Equation (3), it is also fixed at the center of mass to the wall at zero velocity.

### 3.5. Samples

Table 1 lists the peptides used in this study. Pulling through the membrane bilayer of these peptides was investigated. These peptides correspond to the predicted amyloidogenic regions of *E. coli* and *T. thermophilus* ribosomal S1 proteins. Our previous studies have shown that these peptides are capable of forming fibrils and may have antibacterial potential [23].

Peptide 1 was added to the test samples to determine how the size of the peptide is related to the ability to pass through the membrane.

Monopeptides consisting of the same 10 amino acids were selected to study the effect of specific amino acid residues on the force response that occurs during the insertion of the peptide.

The phospholipid was taken as a test sample to better understand the mechanism of resistance to a foreign molecule. An arbitrary phospholipid was selected in the hydrated bilayer. The same force was applied to it as to the peptides, providing movement at a constant speed. Unlike a peptide, a phospholipid is already inside the membrane. Therefore, there is a fundamental difference in which direction to pull it—outside the membrane or inside it. For a better understanding of the membrane resistance mechanism, force pulling was carried out both inside the membrane and outside it (lipid direct pulling and lipid reverse pulling experiments, respectively).

The difference between our work and that done in [37] is that we built peptides with a linear (rather than α-helical) conformation.

The authors of work [37] inserted peptide molecules into the membrane using a molecular constructor (rather than an MD manipulator), and then removed overlapping peptides. The difference between our work and the above is that we studied the penetration of the peptide through the cell membrane using dynamics. This method is called the MD-manipulator method. The molecular constructor uses only static structures for preparation, but using our method, we can see the process of peptide penetration.

## 4. Conclusions

In this work, steered MD simulations of the penetration of amyloidogenic peptides with AMP potential through the model lipid bilayer was carried out to explain the possible mechanism of this process. The force peak corresponds to the entry of the peptide into the membrane. The pulling force per atom is always lower than the force applied to the center of mass, due to the fact that when pulling out the atom, the peptide is pulled into a line; therefore, the area of resistance to the peptide is smaller. Among the amyloidogenic peptides, we identified P2 peptide (EMEVVVLNID from the fifth domain *T. thermophilus* ribosomal S1 protein), the force reaction of which was greater than that of the others. Penetration through the membrane of a hybrid P1 peptide containing CPP (a fragment of the HIV-1 TAT peptide) is easier than CPP and decaarginine (poly (R)) alone. The maximum resistance force depends at least on two factors: (1) partial charges of the atoms; (2) the amino acid resistance area. The best correlation between the maximum reaction force of the membrane and the calculated parameters corresponds to the instability index. It should be noted that any computational experiments serve only as a hint when conducting laboratory experiments; and our computational experiments are no exception.

## Figures and Tables

**Figure 1 ijms-23-05997-f001:**
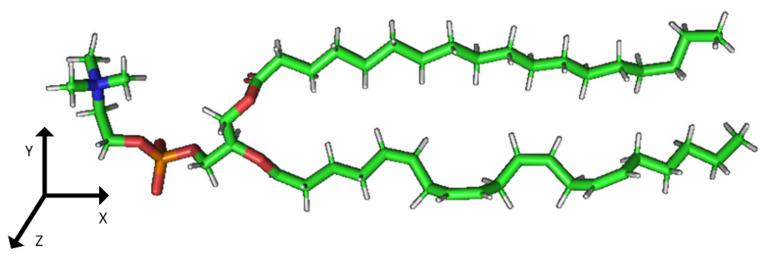
Molecule of 1-palmitoyl-2-oleoyl-sn-glycero-3-phosphatidylcholine (POPC).

**Figure 2 ijms-23-05997-f002:**
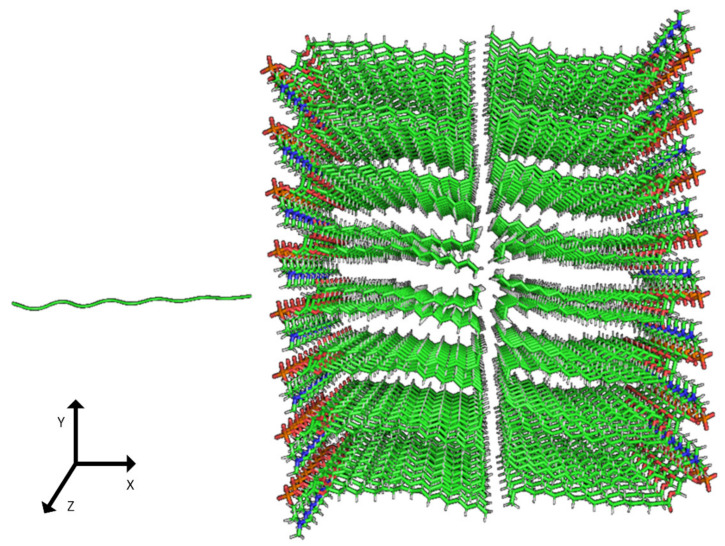
Phospholipid bilayer consisting of 8 × 8 × 2 = 128 molecules of POPC. The addition of the peptide was completed perpendicular to the plane of the membrane.

**Figure 3 ijms-23-05997-f003:**
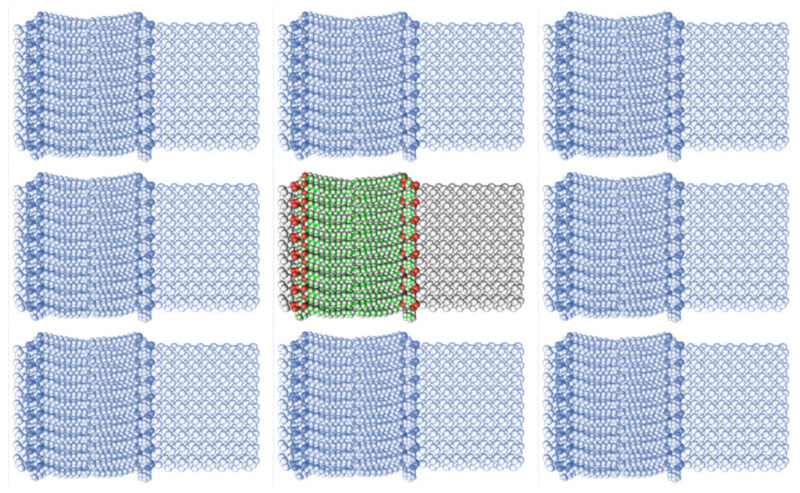
A system (colored) consisting of a phospholipid bilayer, water molecules, and a peptide inside water (not visible), as well as its images (monochrome) under periodic boundary conditions. The distance between the systems is introduced specifically for better understanding.

**Figure 4 ijms-23-05997-f004:**
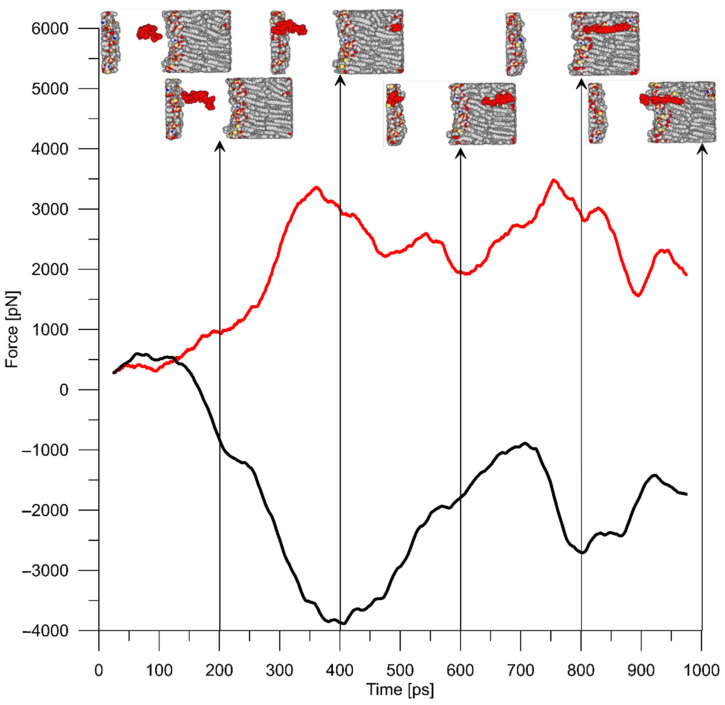
Typical force reaction of P2 peptide (red) and a membrane (black) when the peptide is pulled by the atom at a constant rate of 0.1 Å/ps. Snapshots of this process are presented at different time points: 0, 200, 400, 600, 800, and 1000 ps.

**Figure 5 ijms-23-05997-f005:**
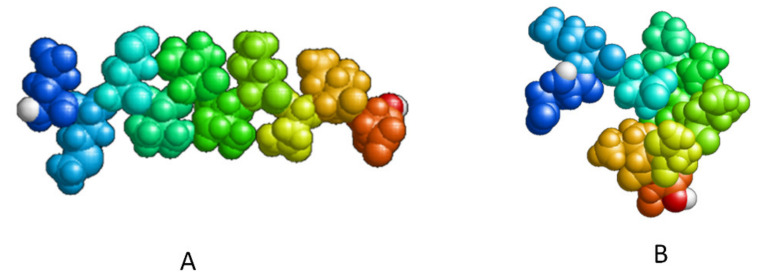
Compactization of P2 peptide (EMEVVVLNID) during relaxation: (**A**) the peptide after construction; (**B**) the peptide after relaxation in an aqueous environment after 1 ns.

**Figure 6 ijms-23-05997-f006:**
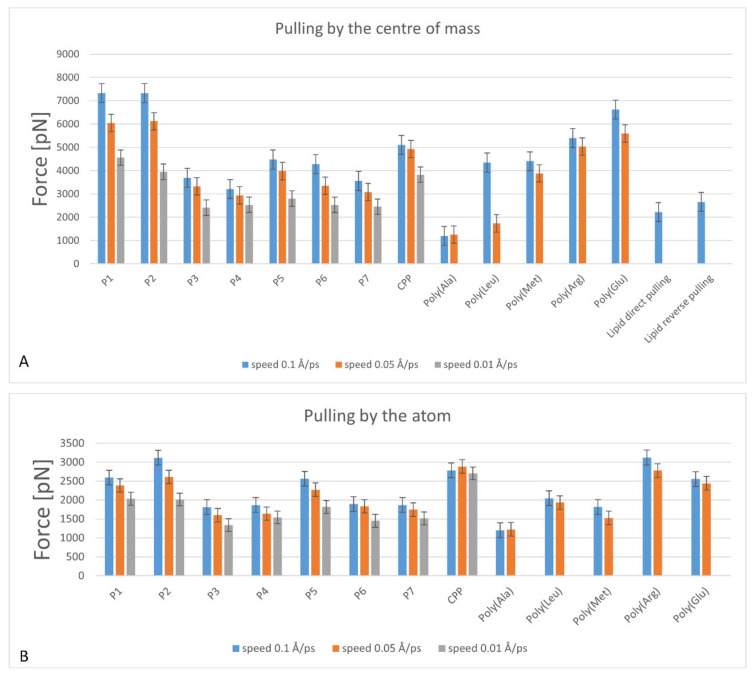
Maximum force statistics from all experiments of pulling peptides through the membrane: (**A**) by the center of mass, (**B**) by the terminal atom. Every column is an averaging of 8 implementations.

**Table 1 ijms-23-05997-t001:** List of amyloidogenic peptides with antibacterial potential from *Escherichia coli* (*E. coli*) and *Thermus thermophilus* (*T. thermophilus*).

Amino Acid Sequence, Localization in Protein and Domain (D), Organism, MW *, Pi ** 3D Structures of the Peptides in the Extended Conformation	Localization in 3D Structure	The Ability of a Peptide to Form Amyloid Fibrils According to the Fluorescence Intensity of Thioflavin T and Electron Microscopy under Condition of 50 mM TrisHCl, pH 7.5; 150 mM NaCl, 1–8 h of Incubation [24]
P1, Peptide 1, RKKRRQRRRGGAGVTDFGVFVEI (R23I) ^T^ VTDFGVFVEI (V10I) ^T^; this fragment is colored in the 3D structure (391–400 a.a.), D5, *T. thermophilus* 2689.1, 12	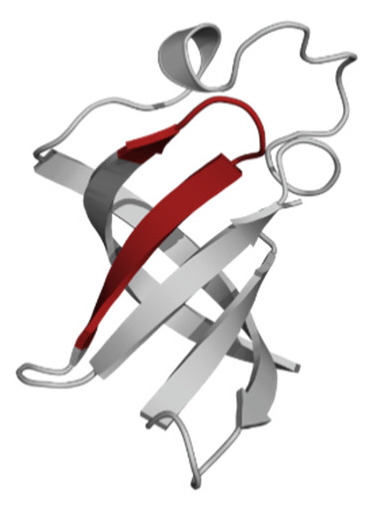	According to the data of fluorescence spectroscopy: the initial low intensity of ThT fluorescence (up to 2 relative units) does not change significantly, which indicates the absence of amyloids in the preparation of the R23I peptide. Aggregates of different sizes and aggregates of such aggregates were observed using electron microscopy (EM).
P2, Peptide 2, EMEVVVLNID (E10D) ^T^, (430–439), D5, *T. thermophilus*, 1160.3, 3.4 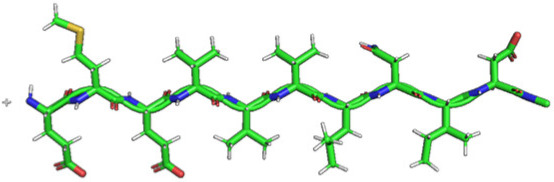	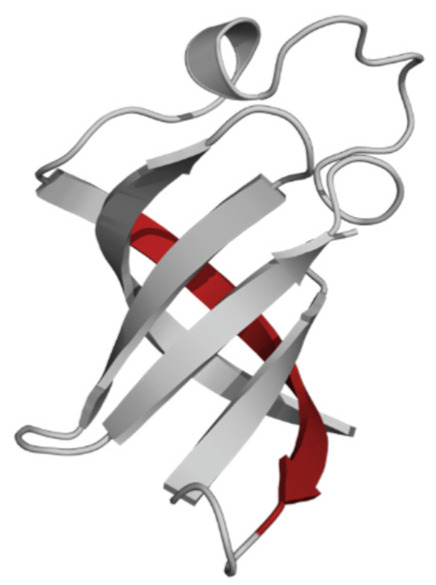	The ThT fluorescence spectrum does not allow for revealing a characteristic increase in fluorescence intensity for ThT bound to amyloid aggregates. According to EM data, the peptide does not form fibrils. Aggregates of different sizes are visible. According to EM data, the peptide forms polymers in the form of short films with a certain height under conditions of 20% acetic acid (pH 2), 150 mM NaCl, 37 °C, incubation time 3 h. Such polymers are collected in aggregates.
P3, Peptide 3, DFGVFVNLG (D9G) ^T^, (221–229 a.a.), D3, *T. thermophilus* 967.1, 3.8 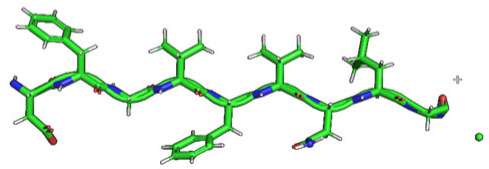	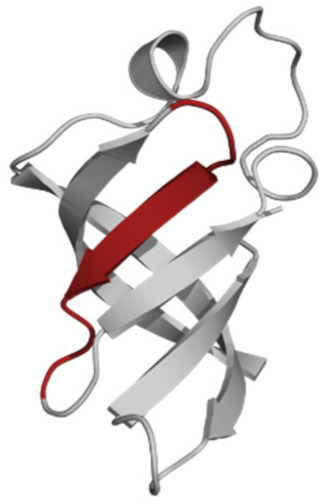	According to the data of fluorescence spectroscopy, an increase in the ThT fluorescence intensity, which increases to 1000 relative units, indicates the presence of amyloids in the peptide preparation. According to EM data, the peptide forms fibrils of different morphology and length.
P4, Peptide 4, IVRGVVVAID (I10D) ^E^, (23–32 a.a.), D1, *E. coli* 1040.3, 6.3 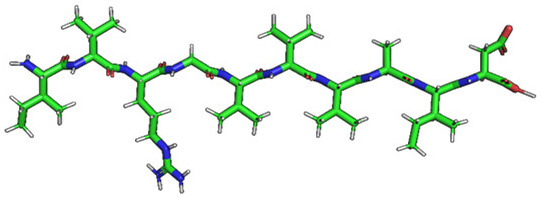	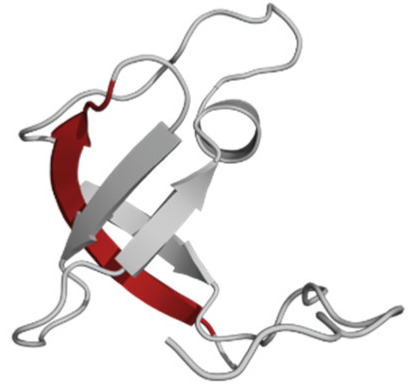	According to the data of fluorescence spectroscopy, the initial high intensity of ThT fluorescence, which decreases to 40 relative units, indicates the presence of amyloids in the peptide preparation. According to EM data, the peptide forms fibrils in the form of small bundles, or spherulins.
P5, Peptide 5, DEITVKVLKF (D10F) ^E^, 3 (239–248 a.a.), D3, *E. coli* 1191.4, 6.3 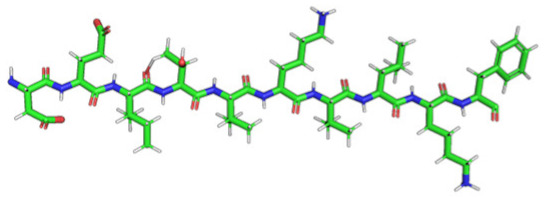	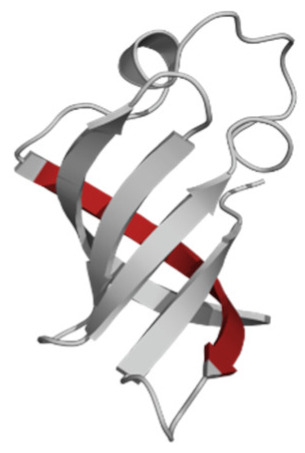	According to the fluorescence spectroscopy data, an increase in the ThT fluorescence intensity, which increases to 60 relative units, indicates the presence of amyloids in the peptide preparation. According to EM data, the peptide forms pre-fibrils. An increase in incubation time leads to the formation of mature fibrils.
P6, Peptide 6, VVEGTVVEVT (V10T) ^T^, (211–220 a.a), D3, *T. thermophilus* 1031.2, 3.5 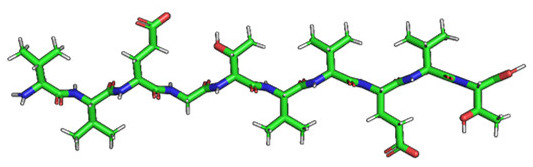	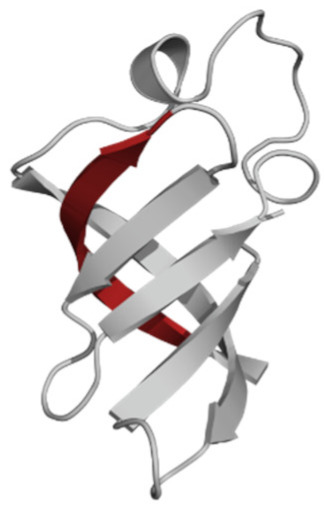	According to the fluorescence spectroscopy data, the initial intensity of ThT fluorescence, which decreases from 12 to 4 relative units, indicates the presence of amyloids (which tend to disaggregate) in the peptide preparation. According to EM data: the peptide forms fibrils in the form of small bundles, which are rare. There are many short pre-fibrils/fibrils.
P7, Peptide 7, VTDFGVFVEI (V10I) ^T^, (391–400 a.a.), D5, *T. thermophilus* 1125.3, 3.5 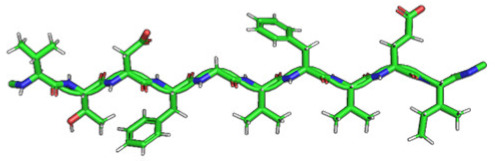	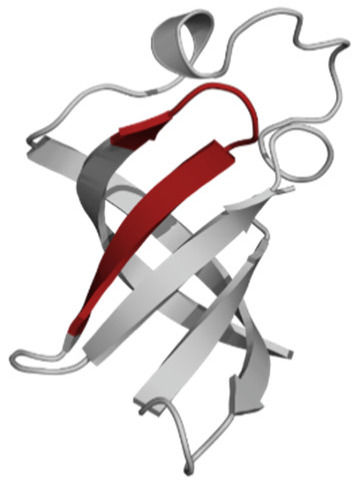	According to the fluorescence spectroscopy data, the initial high intensity of ThT fluorescence, which increases with time, indicates the presence of amyloids in the peptide preparation. According to EM data, the peptide forms fibrils in the form of small bundles and ribbons, which are rare. There are many short pre-fibrils/fibrils.

* and **—molecular weight and isoelectric point were calculated by the ExPASy server [25,26]. ^T^—peptide from the sequence of the ribosomal S1 protein from *T. thermophilus*. ^E^—peptide from the sequence of the ribosomal S1 protein from *E. coli*. 3D structures of the S1 domains from *E. coli*. Experimentally studied amyloidogenic regions (position and amino acid sequence are given) are highlighted with red color. Domain 1—PDB (Protein Data Bank) code: 2MFI; domain 4—PDB code: 2KHI; domain 5—PDB code: 5XQ5. The 3D structure of domain 3 was predicted using the Robetta server. The 3D structures of domains 3 and 5 of the ribosomal S1 protein from *T. thermophilus* were predicted using the Robetta server.

**Table 2 ijms-23-05997-t002:** The values of the maximum peak force averaged over 8 realizations when the peptide is pulled over the center of mass/pulled by the terminal atom.

Peptide	Velocity 0.1 Å/ps	Velocity 0.05 Å/ps	Velocity 0.01 Å/ps
P1	7328/2595	6049/2388	4561/2035
P2	7329/3119	6123/2613	3955/2017
P3	3689/1813	3325/1602	2414/1338
P4	3209/1869	2944/1642	2532/1541
P5	4476/2785	3983/2886	2800/2704
P6	4277/1897	3357/1837	2533/1454
P7	3559/1870	3086/1748	2454/1518
CPP	5107/2785	4934/2886	3828/2704
Poly (Ala)	1201/1201	1258/1226	–
Poly (Leu)	4347/2048	1735/1936	–
Poly (Met)	4405/1819	3880/1529	–
Poly (Arg)	5402/3124	5040/2779	–
Poly (Glu)	6621/2557	5594/2444	–
Lipid direct pulling	2218/-	–	–
Lipid reverse pulling	2655/-	–	–

## Data Availability

The data presented in this study are openly available on the server: http://lmd.impb.ru/protres_membrane.

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
