# Peer review of "Is It Possible to Find an Antimicrobial Peptide That Passes the Membrane Bilayer with Minimal Force Resistance? An Attempt at a Predictive Approach by Molecular Dynamics Simulation"

_ijms, 2022, doi:10.3390/ijms23115997_

Round 1

Reviewer 1 Report

the manuscript consists of extensive molecular dynamics work to determine aspects of a peptide that facilitate membrane crossing.
The two main problems are of method and merit.
In method because the experimental part is not adequately described. E.g., the software employed PUMA is not properly referenced and the two references used point to two inadequate papers predating PUMA development by many years. The software itself is not available and therefore I could not verify its functioning.

In the method because a scientific paper must not consist in a collection of data but must help to understand a phenomenon. For example in the conclusions the authors write: "... It can be assumed that small peptides can penetrate the membrane bilayer much more easily...." I would say that it is not a great discovery to say that a small object crosses a membrane much more easily than a large one. 
Overall the work is superficial and not suitable for publication.

Reviewer 2 Report

The following questions are listed here to be addressed:

  1. It would be good to have a figure that shows a few simulation snapshots during the course of the pulling.  
  2. The authors mentioned that the difference between the maximum force in the independent simulations is minimal, but it would be good to have error bars for Fig. 6.
  3. The author mentioned that small peptides can penetrate the membrane bilayer much more easily, but P2 and P3-P7 doesn’t have an obvious size difference. Could author provide other mechanistic explanations for the finding that P2 experiences the largest force response during pulling among all peptides? Is there any correlation between the membrane-penetration ability and the amyloid-forming propensity?

Minor comments: The point of showing the periodic images of the lipid membrane is not clear. Fig.2 and Fig. 3 could be combined into one figure without showing the periodic images.

Reviewer 3 Report

Comment:

The authors conducted molecular dynamics (MD) simulation to understand lipid membrane penetration of amyloidogenic peptides with antimicrobial potential and poly-peptides with various sequences. In the MD method, a peptide molecule was pulled across the membrane composed of 1-palmitoyl-2-oleoyl-sn-glycero-3-phosphatidylchoine (POPC), and the force resistance measured during this process was used as an indicator for the membrane penetration ability of peptides, in which a lesser force resistance for a peptide means its higher membrane penetration efficiency. The MD method seems to be unique, but its validity is missing at this stage. I recommend publishing this article in the International Journal of Molecular Sciences after the authors address the below points.

  • Should clarify whether the MD method used is originally introduced in this study or not. There is the description “a method for preparing a membrane phospholipid bilayer was proposed” at lines 16–17 in the Abstract section, but I could not see related description in the main text. If the method is newly introduced, the authors should validate their method or discuss its validity by referring to/comparing with relevant previous experimental results. If it is an established method, should provide references/citations with sentences somewhere in the manuscript.

If the authors will validate their method, it is possible to use representative arginine-rich cell-penetrating peptides, such as Tat, polyarginine, Rev, and so on, which have been known to penetrate lipid membranes. The authors indeed examined the membrane penetration of decaarginine (R10), but among the peptides tested in this study, the force resistance for R10 is relatively high despite R10 being a cell-penetrating peptide. However, I could not see any discussion about this. In addition, it is known that the addition of an arginine-rich cell-penetrating aminoacid sequence enhances membrane penetration ability, but it is not the case in this study (P1 vs P7). These results are inconsistent with the knowledge obtained by experimental studies in this field. The authors should discuss these points while referring to previous studies.

  • I was wondering what was the motivation of the authors to set the POPC membrane. Please explain this point. Did the author intend to investigate the membrane penetration in mammalian cells of the peptides with antimicrobial potential? The lipid membrane in this study was exclusively composed of POPC molecules probably because this phospholipid is predominant species in the mammal plasma lipid membrane. Nicer would be explaining the reason (simplifying the system or decreasing the simulation cost?) while referring to mammalian plasma cell membranes being composed of not only phosphatidylcholine (neutrally charged), but also phosphatidylethanolamine (having a negative curvature of molecule), phosphatidylserine (negatively charged), cholesterol (modulating membrane fluidity), and so on.
  • Should discuss major limitations of the MD method used. This point is eventually associated with my comments 1 and 2.
  • Should discuss how the force resistance varies depending on the peptide sequence. What is the driving force for the membrane penetration of peptides in the POPC membrane in the authors’ MD system? I speculated that it is most likely explained by the hydrophobic interaction of a peptide with the POPC membrane because peptides with higher hydrophobicity appears to be advantageous for lipid membrane penetration (Table 2). Adding more discussions regarding this will be helpful for readers.
  • Table 2, Figure 4, and Figure 6: should discuss the degree of forces measured in relation to previous experimental results. For example, are the values of the maximum peak force relevant when compared with previous experimental results? Could you give some examples about what phenomenon can be associated with these specific force values?
  • Figure 6 has no error bar. Would you please add standard deviation or error bar to each value in the Figure 6?
  • At lines 138-142, is the compaction of peptides in the MD simulation derived from favorable conformation change of the peptide in response to water environment? Should discuss the driving force to facilitate the conformation change of peptides. I think such explanation would help readers to understand why the force reaction of the system when the peptide is pulled out by the terminal atom is always lower than when it is pulled out by the center of mass (Table 2 and Figure 6) at lines 142-144.

Round 2

Reviewer 1 Report

I did not find the authors' response satisfactory. Citing other articles using a program is, of course, not sufficient for the publication of this paper.

An insight was and is missing. Unfortunately, this paper represents the typical paper that adds data (moreover not verifiable) and not insight.

Author Response

Dear Reviewer,

of course, it is not easy for us to convince you that our work has an interesting result for readers, although we convinced two reviewers of this. The uniqueness of this work lies in the fact that we have not only carried out these calculations since 2018, but a huge amount of work has been done to test the antibacterial activity of about 30 peptides. All peptides were synthesized and tested for antibacterial activity. Among the amyloidogenic peptides from this work, 2 peptides showed such activity. One of the interesting results of this study is that the 10 residue amyloidogenic peptide and its extended peptide with added 9 residue cell-penetrating peptide and 4 residue linker, both with established antimicrobial activity, have the same bilayer resistance force. We found that the best correlation between the maximum membrane reaction force and the calculated parameters corresponds to the instability index (correlation coefficient is above 0.9). All results of this work are posted at the site http://lmd.impb.ru/protres_membrane. We also provide all programs. Our programs have passed all the necessary checks in due time. In the articles that we have included, now some of them have already been removed from citation at the request of the editor, MD simulations have been published, which were made in comparison with experimental data.

Reviewer 2 Report

The authors have addressed most of my comments and I do not have further questions.

Author Response

Thank you.

Reviewer 3 Report

As long as I know, the MD simulation related the membrane penetration of peptides is typically conducted in terms of free energy as reported in J. Am Chem. Soc. 2014, 136, 17459 and PNAS 2018, 115, 11923. The authors' MD simulation is "actively" pulling the peptide, and this approach seems to be unique, providing new insight into this field. I personally expect that the authors provide their method with the scientific community broadly, which facilitate the understanding the mechanism for the membrane penetration of peptides. Now that the manuscript is improved, I hope that the editor decides to publish the article in the International Journal of Molecular Sciences.

Author Response

Thank you.